# Ethical Leadership and Young University Teachers’ Work Engagement: A Moderated Mediation Model

**DOI:** 10.3390/ijerph17010021

**Published:** 2019-12-18

**Authors:** Jianji Zeng, Guangyi Xu

**Affiliations:** 1School of Medical Business, Guangdong Pharmaceutical University, Guangzhou 510006, China; zengjianji2006@163.com; 2School of Business Administration, South China University of Technology, Guangzhou 510640, China

**Keywords:** ethical leadership, organizational trust, work engagement, S–S guanxi

## Abstract

This paper aims to examine the mediating role of organizational trust in the relationship between ethical leadership and young teachers’ work engagement, and the moderating effect of supervisor–subordinate (S–S) guanxi. S–S guanxi is a special interpersonal relationship in Chinese organizations. The sample in this study comprises 205 young teachers from 15 Chinese universities. The results reveal that organizational trust mediates the relationship between ethical leadership and young teachers’ work engagement. Moreover, S–S guanxi strengthens the positive relationship between organizational trust and young teachers’ work engagement, and the indirect effect of ethical leadership on young teachers’ work engagement through organizational trust. Based upon these findings, several theoretical and practical implications are discussed.

## 1. Introduction

Leadership is a popular research topic, both in business and nonprofit organizations, and leaders are always expected to go beyond the role of managers to run the organization effectively. Recently, numerous ethical scandals in the top management of enterprises have sparked discussions about the ethical aspects of leaders [1]. However, the lack of ethics is not limited to those within business organizations, but also occurs in educators, politicians and other professionals [2]. There is an undeniable fact that values and ethics are important elements of leadership [3], as leaders need to address some ethical dilemmas during the work process. Previous studies have suggested that ethical leadership is strongly associated with employees’ work-related outcomes, such as job satisfaction [4,5], organizational citizenship behavior [6] and voice behavior [7]. 

Although a large number of studies have discussed the impacts of ethical leadership on their subordinates, employees’ work engagement as a key determinant of success for organizations [8] has not been widely concerned [9,10], especially in nonprofit organizations. Such an omission is surprising because employees’ work engagement is strongly related to ethical organizational culture [11]. Moreover, the influential mechanism of the relationship is expected to extend further [12,13]. Ng and Feldman [14] argued that employees of ethical leaders will reciprocate with stronger psychological investment (e.g., trust in the leader or the organization), and in turn display more positive work attitudes and behaviors. Furthermore, Greenfield [15] suggested that school is an important institution, establishing social norms and moral standards, and the managers of universities who are regarded as moral agents have a responsibility to make decisions in ethical ways and exercise authority based on ethical standards. 

As we all know, university young teachers who are characterized by being knowledgeable and by high educational backgrounds are pursuing self-realization. It is essential for young teachers to have favorable working experiences and organizational atmospheres. In addition, in the cultural context of China, Chinese ways of thinking and acting are guided by Confucianism, which is based upon harmonious interpersonal relationships and high collectivism [16]. It is thus valuable to investigate how ethical leadership affects young teachers’ work engagement in China, which is different from Western culture.

The purpose of this study is to examine how ethical leadership affects young teachers’ work engagement in Chinese universities. First, we posit that ethical leadership is positively related to young teachers’ work engagement, following prior studies [9,10,17]. That is, young teachers will engage fully in their work to reciprocate ethical leadership behaviors. Given the characteristics of ethical leaders, we believe the positive effect of ethical leadership holds in both business and nonprofit organizations. Furthermore, we argue that organizational trust plays a mediating role in the relationship. Xu et al. [18] argued that ethical leadership behaviors will promote employees’ trust in the organization, which essentially reflects a stable leader–employee relationship [19]. Employees who possess a higher level of trust in their organization tend to develop favorable expectations about organizational actions and decisions [20], and in turn promote the demonstration of dedication and energy in their work [21]. Therefore, we argue that ethical leadership enhances young teachers’ work engagement by fostering their trust in the organization.

Additionally, we further investigate the boundary condition of S–S guanxi in the relationships. Unlike leader–member exchange (LMX), S–S guanxi is a special interpersonal relationship in Chinese organizations, which refers to the personal exchange after work between supervisors and subordinates. S–S guanxi has been found to have significant impacts on employees’ outcomes [22,23]. Employees in high-quality S–S guanxi who acquire more valuable resources will participate effectively and actively in the process of organizational decision-making [24]. We thus posit that S–S guanxi will moderate the positive effect of organizational trust on young teachers’ work engagement.

The contribution of this study mainly includes the following aspects. First, we provide a new explanatory mechanism for how ethical leadership enhances young teachers’ work engagement. Based on the argument proposed by Ng and Feldman [14], the present study verifies the mediating role of organizational trust. Second, this study deepens the understanding of ethical leadership by considering the boundary condition of S–S guanxi. Although a prior study suggested that S–S guanxi can influence supervisory decisions [23], little research has discussed how S–S guanxi affects the effectiveness of ethical leadership. Third, previous studies on ethical leadership concentrated mainly on business organization. With the sample from Chinese universities, these findings would enrich the understanding of ethical leadership in different types of organizations.

## 2. Theoretical Backgrounds and Hypotheses

To understand the positive effects of ethical leadership behaviors on young teachers, we draw upon social exchange theory [25], which has proven useful for understanding the reciprocal relationship between supervisors and subordinates [14]. The norm of reciprocity is the foundation of social exchange theory [26], which suggests that the parties become involved in and maintain an exchange relationship with others under the expectation of getting returns. Blau [25] made a distinction between economic exchange and social exchange. The former mainly involves the exchange of economic resources, which is essentially instrumental. The latter concentrates on the exchange of social emotional resources [27], suggesting a broader investment in the relationship. Therefore, ethical leadership behaviors signaling a social exchange process may explain why subordinates reciprocate with positive attitudes and behaviors.

Based on social exchange theory, we posit that when subordinates are treated in ethical ways by their supervisors, they will reciprocate positively to those ethical leaders [28,29]. Since ethical leaders are always seen as agents of ethical employers [30], subordinates of ethical leaders will tend to develop positive perceptions of the organization (e.g., organizational trust). 

Organizational trust, one of the forms of vertical trust, refers to the individual’s degree of trust in the organization as a whole [31]. Moreover, trust is a critical factor in identifying whether a social exchange relationship exists [32,33]. When employees have a higher-level trust in the organization, they will be inclined to engage in their work [21,34]. Thus, organizational trust may account for how ethical leadership affects young teachers’ work engagement. From a social exchange perspective, we can achieve a coherent picture of the positive influences of ethical supervisory behavior on young teachers.

### 2.1. Ethical Leadership and Work Engagement

Ethical leadership refers to “the demonstration of normatively appropriate conduct through personal actions and interpersonal relationships, and the promotion of such conduct to followers through two-way communication, reinforcement, and decision-making” [1]. The reputation of the ethical leader is the combination of “moral person” and “moral manager” [35]. The former refers to the personal traits of ethical leaders, such as integrity, honesty, and trustworthiness [1]. Ethical leaders tend to do things based on ethical principles and standards in both their professional and personal lives [17]. The latter represents the leader’s proactive efforts to influence followers’ ethical and unethical behavior. Ethical leaders build and instill ethical standards, not only patterning organizational management to meet moral standards, but also holding subordinates accountable to these standards by using the policies of reward and punishment [36]. In terms of social exchange, ethical leadership should be positively associated with employees’ outcomes (e.g., organizational commitment, organizational citizenship behavior and work engagement) [37]. 

Work engagement is a positive, fulfilling, psychological work-related state that involves vigor, dedication and absorption [38,39]. Demirtas [10] argued that work engagement highlights deep involvement in work, which inspires positive motivations toward the organization [40]. Therefore, engaged employees will tend to pursue personal and organizational goals at the same time [41]. According to social exchange theory, employees under ethical leadership are more likely to develop social exchange relationships with their leaders [25] and reciprocate positively to ethical leaders [28,29] as they are treated in ethical ways [37]. Specifically, ethical leaders are characterized by integrity, honesty and trustworthiness [1], which will motivate employees to engage actively in their work [8,42]. Moreover, ethical leaders encourage employees to participate in organizational decision-making via two-way communication, making them understand the organizational goals and expectations for their work role [30,37], which also fosters employee work engagement [42,43]. Brown et al. [1] found that ethical leadership was strongly associated with followers’ dedication, which is a major dimension of work engagement. Indeed, existing studies have also provided support for the positive effect of ethical supervisory behaviors on subordinates’ work engagement [17,44,45]. Therefore, we propose:

**Hypothesis 1** **(H1).**
*Ethical leadership is positively associated with young teachers’ work engagement.*


### 2.2. Mediating Role of Organizational Trust

Although social exchange theory provides a useful framework for understanding positive effects of ethical leadership on employees’ outcomes, the influential mechanism needs to be further explored. Ethical leaders are always viewed as agents of moral employers and principled decision-makers, and employees of ethical leaders are more likely to develop favorable perceptions of their work environment and the organization as a whole [30]. Therefore, it is believed that ethical leadership can predict various organizational outcomes [18], such as organizational commitment [29] and organizational identification [46,47]. In particular, Ng and Feldman [14] argued that trust is the major lynchpin in a meta-analysis of ethical leadership, which can explain why ethical leaders positively affect their subordinates. In other words, employees of ethical leaders will reciprocate with stronger psychological investment (e.g., trust the organization), and in turn show more positive work attitudes and behaviors. Therefore, we propose that organizational trust is a major linkage.

Employees’ trust in their organization is a necessary element to establish a stable employee–organization relationship [19]. This argument is similar to social exchange theory [25], which emphasizes the importance of trust in the long-term employment relationship. Employees will tend to trust in their organization when they build exchange relationships within the organization. Before deciding whether to trust in the organization or not, employees will constantly observe the work environment and organizational atmosphere [48]. In particular, leadership has an important influence on employees’ perception of trusting of the organization [49,50]. Tan and Tan [51] noted whether employees trust in the organization depends largely on their interaction with their supervisors. Wong et al. [52] also found that employees’ trust in their supervisors is strongly associated with their organizational trust.

Since ethical leaders are responsible for managing employment relationships, it is natural for employees to view ethical leadership behaviors as being rooted in the organization, which in turn will enhance their trust in the organization. Specifically, ethical leaders always display concern and care for employees in name the of the organization [53]. They also make principled and fair decisions that take employees’ needs into account [37]. In addition, employees under ethical leadership are encouraged to participate in the process of organizational decision-making [54]. Prior studies also suggested that ethical leadership behaviors are strongly associated with employees’ trust in management [55] and the development of a trusting organizational atmosphere [56].

As discussed above, organizational trust reflects an individual’s positive expectation of their organization, while work engagement indicates their subsequent involvement with their work, implying the potential influence of organizational trust on employees’ work engagement [21]. Gill and Amarjit [57] argued that employees will dedicate themselves to their work when they enjoy trusting in the organization. The core value of organizational trust can help employees maintain energy and creativity [58], and they will be absorbed in everything the organization does [59]. In addition, organizational trust will increase employees’ sense of attachment to the organization, and thus promote their willingness to work hard. Previous studies have also suggested that organizational trust is positively associated with employees’ work engagement [21,60,61].

In summary, we argue that ethical leadership will promote young teachers’ organizational trust, which in turn fosters their work engagement. Therefore, we propose: 

**Hypothesis 2** **(H2).**
*Organizational trust mediates the positive effect of ethical leadership on young teachers’ work engagement.*


### 2.3. Moderating Effects of S-S Guanxi

S–S guanxi is similar to LMX, in that both emphasize the importance of the quality of the relationship [23], and these two concepts are derived from social exchange theory [25]. LMX is limited to work-related exchanges in the workplace, while S–S guanxi is established by informal interactions after work, and involves a wide range of social emotional exchanges between supervisors and subordinates [62,63]. S–S guanxi is a link based on interests, emotions and identity obligations [24], which is a special interpersonal relationship in Chinese organization [63]. Chen et al. [64] further noted that although S–S guanxi is based upon social exchange theory, the parties involved in S-S guanxi have different obligations and unequal rights. The weaker party will obtain more favor in this exchange relationship. Existing studies have suggested that S–S guanxi helps subordinates to obtain more promotional opportunities [23], promote their career [22,65] and socialize into the organization [66]. Consequently, subordinates tend to trust in their supervisors [64,67], develop favorable perceptions of the organization [68], and demonstrate more positive work-related behaviors. From the social exchange perspective, S–S guanxi may act as a moderator between organizational trust and work engagement.

Drawing on social exchange theory, in high-quality S–S guanxi, supervisors have more confidence in their subordinates, who are thus more likely to acquire favors from their supervisors, such as valuable information or additional resources [24]. Armed with these resources, subordinates can provide more valuable suggestions in the process of organizational decision-making. 

They will be more cognizant of the goals and values of the organization as long as their suggestions are accepted and adopted [69]. Subordinates in this context will tend to reciprocate their supervisors’ support by actively engaging in their work. In addition, the guanxi offers the benefit of solidarity to subordinates, which indicates solid interpersonal trust among the guanxi network [24] and the development of a favorable organizational trusting atmosphere. Based on the social exchange perspective, we propose that high-quality S–S guanxi may strengthen the positive effect of organizational trust on young teachers’ work engagement.

In contrast, subordinates in low-quality S–S guanxi are always considered as the “out-group” by their supervisors [70]. They are unable to obtain necessary favors from their supervisors, and their contributions are hardly recognized [71]. Besides, employees will tend to avoid disagreements with their leaders, and accept directions from those with authority without question [72]. Low-quality S–S guanxi is thus characterized by a low trust level and infrequent interactions between supervisors and subordinates. Therefore, low-quality S–S guanxi may lead to subordinates having no confidence in the organization, which will weaken the positive impact of organizational trust upon young teachers’ work engagement. Therefore, we propose:

**Hypothesis 3** **(H3).**
*S–S guanxi positively moderates the relationship between organizational trust and young teachers’ work engagement, such that this relationship is much stronger with high- rather than low-quality S–S guanxi.*


Based on H1–H3, we further propose a moderated mediation model, that is, S–S guanxi moderates the indirect effect of ethical leadership on young teachers’ work engagement via organizational trust. Thus, we propose: 

**Hypothesis 4** **(H4).**
*S–S guanxi moderates the indirect effect of ethical leadership on young teachers’ work engagement via organizational trust, such that the indirect effect is more positive with high- rather than low-quality S–S guanxi.*


According to the arguments above, we propose the following theoretical model (Figure 1).

## 3. Methodology

### 3.1. Participants and Procedures

The sample of this study comprised 205 young teachers from 15 Chinese universities. All questionnaires in this study were completed anonymously, and the purpose of this study was explained to all respondents. A total of 300 questionnaires were sent to the participants. We received 225 completed responses, and 20 questionnaires that did not meet the requirements of this study were eliminated, resulting in 205 valid, obtained questionnaires (68% valid response rate).

Among the valid sample, 50.7% were male, 48.8% were female, and one respondent did not report their gender. In terms of age, 2% of them are 25 or younger, 16.6% are from 26 to 30 years old, 43.9% are from 31 to 35 years old, 25.9% are 35 to 40 years old, and 11.7% are from 41 to 45 years old.

In terms of education level, 1% had degree, 12.7% had a bachelor’s degree, 42.9% had a master’s degree, 42.4% had a doctoral degree and 1% did not report their education level. In terms of professional titles, 71.7% were lecturers or below, 24.8% were associate professors, 2% were professors and 1.5% did not report their title.

### 3.2. Measures 

To ensure the reliability of the measurement tools, the scales in this study were taken from established studies. We conducted a strict, two-way translation process, such that the measurement scales were translated into Chinese and then translated back into English. In addition, to ensure that all the items are applicable to the research context, some minor modifications were made, following suggestions from three professors in a relevant research field. All the measures were scored with 7-point Likert scales, ranging from 1 (“strongly disagree”) to 7 (“strongly agree”). All measurement items were presented in the Appendix A. 

#### 3.2.1. Ethical Leadership

This construct was measured using Zheng et al.’s [73] scale, which has five items in total. Typical questions include “My supervisor is a decent person who do not seek personal interests” and “My supervisor is a good example of our life and work”. The five-item scale reliability was 0.95.

#### 3.2.2. Organizational Trust

Organizational trust was measured using Robinson’s [74] seven-item scale. Typical questions include “My school is always honest and trustworthy” and “In general, I believe the school’s motives and intentions are good”. The reliability of these seven items was 0.96.

#### 3.2.3. Work Engagement

Work engagement was measured using Kanuage’s [75] scale, including ten items. Typical questions included “The most important things that happen to me involve my present job” and “I am very much involved personally in my job”. The reliability of these ten items was 0.92.

#### 3.2.4. S–S guanxi 

To measure S–S guanxi, a six-item scale developed by Law et al. [23] was used. Typical questions included “I call or visit my supervisor after work or on vacation” and “I actively share my thoughts, problems, needs or feelings with my supervisor”. The reliability of these six items was 0.86.

#### 3.2.5. Control Variables 

In this study, several demographic variables were selected as control variables, including gender, age, education level and professional title. These variables may influence the relationships among ethical leadership, organizational trust, work engagement and S–S guanxi. Gender was set as a dummy variable. Education level and professional title were set as an ordinal variable. Age was set as the median of each category. The coding of these control variables is presented in Table 1.

## 4. Results

### 4.1. Common Method Variance Tests

This study examines the effects of ethical leadership from the perspective of individual psychological perception. The data collected were self-reported, which may cause common method variance (CMV). Thus, we adopted survey procedure control and statistical analysis control to reduce the CMV of the data. First, to eliminate respondents’ concerns, the questionnaires were filled in anonymously. Meanwhile, reverse questions were set in the questionnaire to reduce the potential tendency consistency. Second, Harman’s single-factor analysis was also used to test the CMV of all the items of each variable. 

The result of exploratory factor analysis (EFA) shows that the first variable could only explain 38.97% of all measured variation before rotation, which is less than the recommended value of 50%, indicating that the CMV was within the acceptable range in this study.

### 4.2. Reliability and Validity

In this study, we used SPSS 23.0 statistical software (IBM, Armonk, NY, USA) to test the reliability of the collected data. The results in Table 1 show that the Cronbach’s alpha coefficients of the scales of ethical leadership, organizational trust, work engagement and S–S guanxi were 0.95, 0.96, 0.92 and 0.86, respectively, all of which are greater than 0.8. These outcomes indicate that the internal consistency of each variable is good, and thus the reliability is good. In terms of validity, principal component analysis (PCA) and varimax rotation were used to conduct EFA on the scale. The KMO values of each variable were 0.88, 0.93, 0.92 and 0.81, respectively. The cumulative variance contribution rates were 83.03%, 78.97%, 62.56% and 64.70%, respectively. The factor loads of each item in the scale were more than 0.6. In conclusion, the data in this study has good reliability and validity. 

### 4.3. Descriptive Statistics

As illustrated in Table 1, the mean values of ethical leadership, organizational trust, work engagement and S–S guanxi were 4.38, 4.55, 5.54 and 3.38, respectively. Ethical leadership was significantly correlated with organizational trust (r = 0.54, *p* < 0.01), work engagement (r = 0.36, *p* < 0.01), and S–S guanxi (r = 0.38, *p* < 0.01). Organizational trust was significantly correlated with work engagement (r = 0.46, *p* < 0.01) and S–S guanxi (r = 0.22, *p* < 0.01). There was no significant correlation between work engagement and S–S guanxi (r = 0.01, *p* = 0.05). The correlations among these variables basically confirm the theoretical predictions, which provides preliminary support for the hypotheses of this study. 

### 4.4. Hypotheses Testing

#### 4.4.1. Principal Effect Testing

In this study, hierarchical regression was used to test the effect of ethical leadership on young teachers’ work engagement. First, demographic variables were introduced into the regression model. Second, ethical leadership was added into the regression model (see Table 2). According to model 1, gender, age, educational background and professional title have no significant impact upon work engagement. Model 2 added the independent variable to test whether ethical leadership is associated with work engagement. The results show that the explanatory ability of the model increased to 15.2%. Meanwhile, ethical leadership was significantly and positively related to work engagement (β = 0.378, *p* < 0.01) in model 2. Therefore, Hypothesis 1 was supported.

#### 4.4.2. Mediating Effect of Organizational Trust

The result in model 2 satisfies the first condition of Baron and Kenny’s [76] approach for mediation. Models 9, 3 and 4 represent the second, third and fourth conditions of mediation, respectively. The result in model 9 (β = 0.547, *p* < 0.01) shows that ethical leadership was significantly and positively associated with organizational trust. Similarly, organizational trust was found to be significantly and positively associated with work engagement (β = 0.455, *p* < 0.01) in model 3. The result in model 4 further shows that the significant impact of ethical leadership on work engagement still exists when organizational trust was introduced in the regression of work engagement, and it has a significantly positive impact on work engagement (β = 0.357, *p* < 0.01). These results indicated that organizational trust plays a mediating role. Therefore, hypothesis 2 was supported.

The bootstrapping technique was used to further verify the mediating effects of organizational trust. The results show that ethical leadership’s indirect effect on work engagement through organizational trust was 0.146, with the 95% confidence interval (CI) being [0.082, 0.223] (see in Table 3). Ethical leadership’s direct effect on work engagement through organizational trust is 0.123, with the 95% confidence interval (CI) being [0.016, 0.230]. The two CIs do not contain zero, which accordingly confirms that ethical leadership affects work engagement through organizational trust. Therefore, hypothesis 2 was further supported.

#### 4.4.3. Moderating Effects of S–S guanxi 

In this study, the moderating effect of S–S guanxi occurs after the mediator of organizational trust. That is, the mediating effect of organizational trust works through S–S guanxi. According to the method for testing the moderated mediation proposed by Muller et al. [77], the first step was to test the impact of ethical leadership and S–S guanxi on work engagement. The coefficient of ethical leadership should be significant. 

The result in model 5 shows that the regression coefficient of ethical leadership was indeed significant (β = 0.433, *p* < 0.01). The second step was to test the impact of ethical leadership and S–S guanxi on organizational trust. The coefficient of ethical leadership should be significant. Model 8 shows that the regression coefficient of ethical leadership was indeed significant (β = 0.538, *p* < 0.01). The third step was to test the impact of ethical leadership, S–S guanxi and organizational trust on work engagement. The coefficient of organizational trust should be significant. Model 6 shows that the regression coefficient of organizational trust was indeed significant (β = 0.362, *p* < 0.01). The fourth step was to test the impact of ethical leadership, S–S guanxi, organizational trust and the interaction between organizational trust and S–S guanxi on work engagement. The coefficient of interaction between S–S guanxi and organizational trust should be significant. As seen in model 7, the regression coefficient of the interaction between S–S guanxi and organizational trust was indeed significant (β = 0.151, *p* < 0.05). These results indicated that both hypothesis 3 and hypothesis 4 were supported.

To clearly describe the moderating effects of S–S guanxi, we adopted the methods and procedures developed by Aiken and West [78]. The moderating effects of S–S guanxi at higher (mean +1 standard deviation (SD)) and lower (mean −1 standard deviation) level on organizational trust and young teachers’ work engagement were plotted (see Figure 2). It can be seen from Figure 2 that, compared with low-quality S–S guanxi, high-quality S–S guanxi can strengthen the positive impact of organizational trust on work engagement.

## 5. Conclusions and Discussion

As a moral agent of the organization, an ethical leader plays an important role in organizational management. Based on social exchange theory, this study investigates the influential mechanism and boundary condition of ethical leadership on young teachers’ work engagement in Chinese universities. Specifically, the results reveal that organizational trust serves as a mediator in the relationship between ethical leadership and young teachers’ work engagement. In addition, S–S guanxi is found to positively moderate the relationship between organizational trust and young teachers’ work engagement, and the indirect effect of ethical leadership on work engagement via organizational trust. Although this study was conducted in a Chinese cultural context, which is different from Western culture, the effect of ethical leadership on subordinates’ work engagement is similar to prior studies [9,10]. Several theoretical and practical implications can be found in this study.

### 5.1. Theoretical Implications

First, this study provides a new perspective for understanding the effect of ethical leadership on their subordinates. While previous studies had linked ethical leadership to work engagement [44,79], we go a step further to explore how ethical leadership affects young teachers’ work engagement. This study empirically tests the proposition argued by Brown and Treviño [37] that ethical leadership is related to employees’ outcomes. Specifically, the results reveal that ethical leadership promotes young teachers’ work engagement by fostering their trust in the organization. That is, young teachers of ethical leaders tend to develop trust in the organization, and they in turn actively engage in their work.

Second, this study emphasizes the importance of a trusting relationship, employees–organization. As a moral agent of the organization, the ethical leader represents the interests of the organization [18]. Our results suggest that ethical leadership is a critical source of young teachers’ organizational trust, which is consistent with Whitener et al.’s [80] argument that trustworthy managerial behavior provides a foundation for employees’ perceptions of trusting. When the leader is perceived as ethical and moral, young teachers will tend to build a strong trusting relationship with the organization. 

Third, the present study identifies S–S guanxi as an important boundary condition. Gong et al. [81] argued that the effectiveness of trust can be affected by relational factors. We offer empirical support for this argument by revealing that S–S guanxi moderates the relationship between organizational trust and young teachers’ work engagement. As one type of Chinese social capital, in the context of high-quality S–S guanxi, organizational trust can work better to promote young teachers’ work engagement. 

Additionally, this study focused on young teachers of Chinese universities. Thus far, very few studies had investigated ethical leadership behaviors and the effects on subordinates’ outcomes in nonprofit organizations. Since young teachers possess the characteristics of being knowledgeable and having a high educational background, they strive for self-realization and place more emphasis on ethical leadership behaviors. Interestingly, the findings of this study confirm that ethical leadership also plays a critical role in nonprofit organization. The results suggest that ethical leadership is beneficial and valuable across different types of organizations.

### 5.2. Practical Implications

This study also provides several implications for management. First, this study confirms that ethical leadership is effective in evoking subordinates’ trust in the organization and subsequently enhancing their work engagement. Based on these findings, universities should firmly maintain the rules of ethics, and select managers with high moral qualities who highlight the importance of ethical behaviors. In addition, the managers of universities should proactively demonstrate genuine concern and offer ethical guidance to young teachers. For example, to promote young teachers’ work engagement, the managers should listen to young teachers’ opinions and encourage them to participate in the process of organizational decision-making.

Second, our results suggest that leaders’ behaviors (e.g., ethical or unethical) perceived by subordinates will develop their judgment of whether the organization is trustworthy. To facilitate young teachers’ trust in the organization, not only should leaders be “moral persons” (e.g., characterized as having honesty, trustworthiness and integrity), but also they should be “moral managers” (e.g., making balanced and fair decisions or inculcating ethical principles in the organization) who deal with moral dilemmas in normatively appropriate ways. The managers who are perceived as ethical leaders can build trusting relationships with young teachers and enhance their engagement with their work.

Third, as mentioned earlier, relational factors may impact the effectiveness of trust, and our findings of S–S guanxi support this argument. The results reveal that the quality of S–S guanxi will impact the effectiveness of ethical leadership and organizational trust. Therefore, the managers should be encouraged to establish good personal relationships with young teachers by offering sincere care about their lives, and so on. 

### 5.3. Limitations and Future Research

There are also some limitations in this study. First, the data collected in this study was single-sourced and self-reported, which may raise CMV issues. Although the results of CMV testing were within acceptable thresholds, the use of different data sources is more rigorous in ethical leadership research. For example, to avoid self-report bias, future studies can collect information about ethical leadership from other sources, such as ethical leaders’ peers or supervisors. Besides, the sample in this study was taken only from Chinese universities, which may affect the generalizability of the results to other contexts. Future studies can investigate whether the findings of this study can be found in other cultural settings. Given the findings of this study, we hope to encourage more studies to focus on the research field of ethical leadership, especially in nonprofit organizations.

## Figures and Tables

**Figure 1 ijerph-17-00021-f001:**
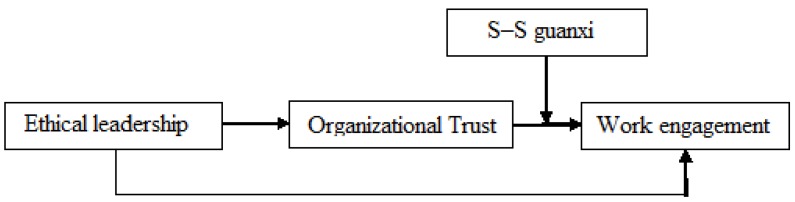
Theoretical model.

**Figure 2 ijerph-17-00021-f002:**
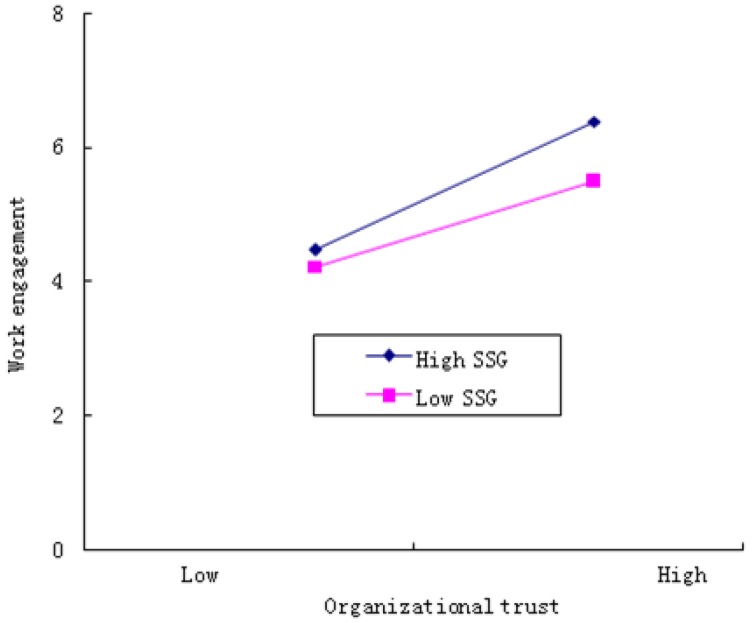
Interactive effects of organizational trust and S–S guanxi on work engagement.

**Table 1 ijerph-17-00021-t001:** Correlation coefficients of each variable and Cronbach’s alpha.

Variables	Mean	SD	1	2	3	4	5	6	7	8
1. Gender	1.53	0.73	N/A							
2. Age	34.44	4.72	−0.14 *	N/A						
3. Education	3.34	0.91	−0.11	0.10	N/A					
4. Title	1.40	1.05	−0.01	0.33 **	0.27 **	N/A				
5. EL	4.38	1.34	0.06	−0.09	0.04	0.07	(0.95)			
6. OT	4.55	1.25	0.05	0.02	0.03	0.05	0.54 **	(0.96)		
7. WE	5.52	0.99	0.08	0.05	−0.01	−0.03	0.36 **	0.46 **	(0.92)	
8. SSG	3.38	1.27	−0.09	−0.04	−0.06	0.08	0.38 **	0.22 **	0.01	(0.86)

Note. N = 205; EL = ethical leadership, OT = Organizational trust, WE = Work engagement, SSG = S–S guanxi. Gender is coded as 1 = male, 2 = female. Age is coded as 23 = 25 years old or younger, 28 = 26–30 years, 33 = 31–35 years, 38 = 36–40 years, 43 = 41–45 years. Education is coded as 1 = associate degree, 2 = bachelor’s degree, 3 = master’s degree, 4 = doctoral degree. Title is coded as 1 = lecturer or below, 2 = associate professor, 3 = professor. ** *p* < 0.01, * *p* < 0.05. Values shown in parentheses are Cronbach’s alpha of latent variables.

**Table 2 ijerph-17-00021-t002:** Results of hierarchical regression analysis.

Variables	Dependent Variable: WE							Mediator: OT	
	M_1_	M_2_	M_3_	M_4_	M_5_	M_6_	M_7_	M_8_	M_9_
Control variables									
Gender	0.091	0.072	0.069	0.065	0.053	0.044	0.034	0.025	0.022
Age	0.079	0.126	0.075	0.099	0.119	0.092	0.085	0.077	0.076
Education	0.013	0.002	0.000	−0.003	−0.014	−0.020	−0.011	0.016	0.013
Title	−0.060	−0.097	−0.077	−0.091	−0.084	−0.077	−0.071	−0.020	−0.017
Research variables									
EL		0.378 **		0.182 **	0.433 **	0.238 **	0.239 **	0.538 **	0.547 **
OT			0.455 **	0.357 **		0.362 **	0.400 **		
SSG					−0.144 *	−0.152 *	−0.137 *	0.023	
OT × SSG							0.151 *		
R^2^	0.013	0.152	0.219	0.242	0.169	0.261	0.282	0.298	0.297
⊿R^2^	−0.007	0.131 **	0.199 **	0.219 **	0.144 **	0.235 **	0.252 **	0.276 **	0.280 **
F	0.651	7.148 **	11.166 **	10.537 **	6.728 **	9.947 **	9.602 **	13.985 **	16.831 **

Note. N = 205; EL = ethical leadership, OT = Organizational trust, WE = Work engagement, SSG = S–S guanxi. ** *p* < 0.01; * *p* < 0.05 (two-tailed test).

**Table 3 ijerph-17-00021-t003:** Bootstrapping analysis results of the mediation effect in organizational trust.

Effect Categories	Effects	SE	95% LLCI	95% ULCI
Indirect effect	0.146	0.035	0.082	0.223
Direct effect	0.123	0.054	0.016	0.230

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
