# Peer review of "Ethical Leadership and Young University Teachers’ Work Engagement: A Moderated Mediation Model"

_ijerph, 2019, doi:10.3390/ijerph17010021_

Round 1

Reviewer 1 Report

I do not understand Mean age of participants 3.29? How is it possible. Mean for education which is categorical data? What corelaction have been used?

Presentation of Alphas – diagonal in the table seems tricky especially they are mixed with correlations for 1-4

The conclusions seem quite obvious

Author Response

Response to Reviewer 1 Comments

First of all, we would like to thank the reviewer and the editor for the positive and constructive comments and suggestion.

Point 1: I do not understand Mean age of participants 3.29? How is it possible. Mean for education which is categorical data? What correlation have been used?

Response 1: Thank you for your comment. In this study, gender was set as dummy variable, and age, education, title were set as ordinal variable. Gender is coded as 1=male, 2=female. Age is coded as 1=25 years old or younger, 2=26–30 years, 3=31–35 years, 4=36–40 years, 5=41–45 years. Education is coded as 1=bachelor’s degree below, 2=bachelor’s degree, 3=master’s degree, 4=doctoral degree. Title is coded as 1=lecturer or below, 2=associate professor, 3=professor. We’ve already revised in the “control variable section” and the footnote of table 1.

Point 2: Presentation of Alphas – diagonal in the table seems tricky especially they are mixed with correlations for 1-4.

Response 2: Thank you for your comment, the diagonal values in table 1 are reliability coefficients. To be honest, the presentation of Alphas – diagonal are easily confused with correlations for 1-4. we have added parentheses to Alphas in diagonal, then the footnote of “The diagonal of the latent variable is Cronbach’ alpha” in table 1 was corrected as “Values shown in parentheses are Cronbach’ alpha of latent variables”.    

Point 3: The conclusions seem quite obvious

Response 3: Thank you for your comment. The purpose of this study is to investigate how and when ethical leadership behavior influences young teacher’s work engagement. We strive to expand the research of ethical leadership. For one thing, this study provides a new perspective for understanding the effect of ethical leadership on their subordinates, and we highlight the importance of trusting relationship employee-organization. For another, S-S guanxi was identified as an important boundary condition in this study, which is a special relationship in Chinese organizations. Additionally, this study focused on young teachers of Chinese universities. As we all know, university young teachers who are highly educated and knowledgeable in their fields are pursuing self-realization. The influence mechanism of ethical leadership on work engagement may be different between university young teachers and enterprise employees. For university young teachers, they may pay more attention to the moral quality of their supervisor. Based on this, we think that the findings of this study have certain theoretical and practical implications.

Reviewer 2 Report

please check the sentence structure and wording of page 3 lines 97-99. specifically the word 'conducts' which is used again in line 103

in table 1 it is not clear why you include categorical variables such as gender and title in the correlations. The control variable section lines 254-256 might be clearer if you explain how you measured age -- continuous variable, ordinal variable, etc.

line 371 should use 'focused' rather than 'focus' and 'studies' rather than 'study'

since the introduction indicated that Chinese culture is different than Western culture you might wish to note in the conclusion section if the results found are different than similar studies in Western cultures

Author Response

Response to Reviewer 2 Comments

First of all, we would like to thank the reviewer and the editor for the positive and constructive comments and suggestion.

Point 1: Please check the sentence structure and wording of page 3 lines 97-99. specifically the word ‘conducts’ which is used again in line 103.

Response 1: Thank you for your comment, we’ve already revised the sentence of  page 3 lines 97-99, and we quote directly the definition of ethical leadership defined by Brown et al. (2005), “the demonstration of normatively appropriate conduct through personal actions and interpersonal relationships, and the promotion of such conduct to followers through two-way communication, reinforcement, and decision-making”(lines 98-100). In addition, the sentence of  “The latter refers to how leaders promote ethical conducts at work” (line 103) was corrected as “The latter represents the leader’s proactive efforts to influence followers’ ethical and unethical behavior” (lines 106-107).

Point 2: In table 1 it is not clear why you include categorical variables such as gender and title in the correlations. The control variable section lines 254-256 might be clearer if you explain how you measured age -- continuous variable, ordinal variable, etc.

Response 2: Thank you for your comment. In this study, gender was set as dummy variable, and age, education, title were set as ordinal variable. Gender is coded as 1=male, 2=female. Age is coded as 1=25 years old or younger, 2=26–30 years, 3=31–35 years, 4=36–40 years, 5=41–45 years. Education is coded as 1=bachelor’s degree below, 2=bachelor’s degree, 3=master’s degree, 4=doctoral degree. Title is coded as 1=lecturer or below, 2=associate professor, 3=professor. We’ve already revised in the “control variable section” and the footnote of table 1.

Point 3: Line 371 should use ‘focused’ rather than ‘focus’ and ‘studies’ rather than ‘study’.

Response 3: We are very sorry for our incorrect writing and we have corrected these writing errors in line 390-391.

Point 4: Since the introduction indicated that Chinese culture is different than Western culture, you might wish to note in the conclusion section if the results found are different than similar studies in Western cultures.

 Response 4: It is really true as reviewer suggested that we should point out whether the results is different from prior studies in Western cultures. We have made a complement in conclusion section (line 364-366), “Although this study was conducted in Chinese cultural context, which is different from western culture, the effect of ethical leadership on subordinates’ work engagement is similar to prior studies” (Chughtai et al. 2015; Demirtas, 2015).

Reference:

Brown, M.E.; Treviño, L.K.; Harrison, D.A.. Ethical leadership: A social learning perspective for construct development and testing. Organ. Behav. Hum. Decis. Process. 2005, 97, 117-134.

Chughtai, A.; Byrne, M.; Flood, B.. Linking ethical leadership to employee well-being: the role of trust in supervisor. J. Bus. Ethics 2015128, 653-663.

Demirtas, O.. Ethical Leadership Influence at Organizations: Evidence from the Field. J. Bus. Ethics 2015, 126, 273-284.

Reviewer 3 Report

The paper is well written and interesting to read. With some minor revisions it will be ready for publication! It was easy to understand the theoretical model from both abstract and introduction. Well done!

Abstract:

Explain with one sentence what S-S guanxi is. For instance use the sentence S-S guanxi "is a special interpersonal relationship in Chinese organizations" from line 173 in the paper.

Methodology: 

Explain in more detail how and by whom (The authors? A profesional translator? etc.) the translation of the scales were made.

I would also like to see the scales (ET five items, OT seven items, WE ten items, and S-s guanxi six items), either in the paper as a table or as a addition at the end. 

Reults

Table 1: number 7 has turned into a 6 in the first column (Varibles).

Line 299: says "increased by 15,2%" change to "increased to 15,2%"

Line 300; 302; 306: you have lost an "l" in model in these lines. Please correct!

Author Response

Response to Reviewer 3 Comments

First of all, we would like to thank the reviewer and the editor for the positive and constructive comments and suggestion.

Point 1: Abstract

Explain with one sentence what S-S guanxi is. For instance use the sentence S-S guanxi "is a special interpersonal relationship in Chinese organizations" from line 173 in the paper.

Response 1: Thank you for your comment. We have supplemented the concept of S-S guanxi in abstract section, “S-S guanxi is a special interpersonal relationship in Chinese organizations” in line 10-11.

Point 2: Methodology

Explain in more detail how and by whom (The authors? A profesional translator? etc.) the translation of the scales were made.

I would also like to see the scales (ET five items, OT seven items, WE ten items, and S-s guanxi six items), either in the paper as a table or as a addition at the end. 

Response 2: Thank you for your comment. We have made a supplement about the translation process of the scales in measure section, “we conducted a strict two-way translation process such that the measurement scales were translated into Chinese and then translated back into English. In addition, to ensure that all the items are applicable to the research context, some minor modifications were made following suggestions from three professors in a relevant research field”.

All scales of in this study (ET five items, OT seven items, WE ten items, and S-s guanxi six items) are presented in the appendix.

Point 3: Reults

Table 1: number 7 has turned into a 6 in the first column (Varibles).

Line 299: says “increased by 15,2%” change to “increased to 15,2%”.

Line 300; 302; 306: you have lost an “l” in model in these lines. Please correct!

Response 3: We are very sorry for our incorrect writing and we have corrected these writing errors

Round 2

Reviewer 1 Report

I am still of the opinion that the presentation of the median age in the form of a median category is unfortunate. It seems better to use class centers or gray numbers. The explanation, however, threw a little light and despite all the certain substantive remarks the results are more understandable.

Author Response

Response to Reviewer 1 Comments

we would like to thank the reviewer and the editor for the positive and constructive comments and suggestion.

Point 1: I am still of the opinion that the presentation of the median age in the form of a median category is unfortunate. It seems better to use class centers or gray numbers. The explanation, however, threw a little light and despite all the certain substantive remarks the results are more understandable.

Response 1: Thank you for your comment. It is really true as reviewer suggested that the presentation of the median age in the form of a median category is inappropriate. According to the review’s suggestions, we have made corresponding modifications. In this study, Age was set as the median of each category. Namely, age is coded as 23=25 years old or younger, 28=26–30 years, 33=31–35 years, 38=36–40 years, 43=41–45 years. After adjusting the age data, we did correlation and regression analysis again. The results show that the findings of this study have not been changed after adjusting the age valuable. We have revised the changes in table1.
